

# Large pelvic tubercle in orangutans relates to the adductor longus muscle

Brian M. Shearer[1,2,3], Magdalena Muchlinski[4] and
Ashley S. Hammond[2,5]

[1] Department of Cell Biology, New York University School of Medicine, New York, NY, USA
[2] New York Consortium in Evolutionary Primatology, American Museum of Natural History, New York, NY, USA
[3] Department of Anthropology, City University of New York, Graduate School and University Center, New York, NY, USA
[4] Anatomical Services Center, Oregon Health Sciences University, Portland, OR, USA
[5] Division of Anthropology, American Museum of Natural History, New York, NY, USA

Corresponding author
Brian M. Shearer,
bian.shearer@nyumc.org

## ABSTRACT

Orangutan pelves commonly exhibit a large, projecting tubercle in the iliopubic region, historically assumed to homologous to the pubic tubercle in humans. However, it is not clear whether this tubercle is a unique feature of *Pongo*, or if it is anatomically homologous with the human pubic tubercle when considered as a soft tissue attachment point. To clarify this issue, we dissected orangutan and other ape cadaveric specimens to evaluate the pelvic brim soft tissues and how they may relate to the tubercle (when present). We additionally conducted a broad osteological survey of pelvic brim morphology across 28 primate genera (*n* = 294 specimens) to document the presence of the tubercle in primate pelves. Cadaveric dissections revealed that the tubercle is exclusively associated with the proximal attachment of the adductor longus muscle tendon in orangutans. Our osteological survey confirms that the tubercle is both constantly present and very prominent in orangutans. We observed that the tubercle is consistently situated along the pectineal line, lateral to where the pubic tubercle in humans is found, thereby making its structural homology unlikely. The osteological survey documented the tubercle at polymorphic frequencies in all hominoid taxa, though generally less protuberant than observed in *Pongo*. We argue that this further excludes its possibility of homology with the pubic tubercle, and that it may therefore be more appropriately be considered an adductor longus tubercle. We discuss possible functional and phylogenetic implications for this feature.

## INTRODUCTION

The orangutan (*Lacépède, 1799*: *Pongo pygmaeus*, *Pongo abelii*) is unusual among primates in that it displays a large, flaring tubercle in the area of the iliopubic junction (Figs. 1 and 2). Early comparative anatomists do not agree on the identity of this structure, with some descriptions suggesting that it could be a laterally positioned pubic tubercle (i.e., homologous to the medial attachment site of the inguinal ligament in humans;

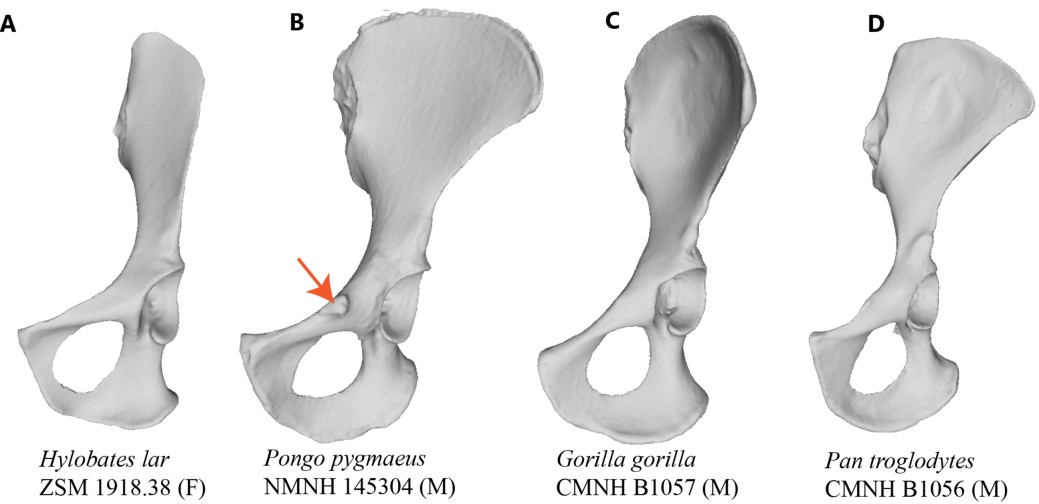

A  *Hylobates lar*
ZSM 1918.38 (F)

B  *Pongo pygmaeus*
NMNH 145304 (M)

C  *Gorilla gorilla*
CMNH B1057 (M)

D  *Pan troglodytes*
CMNH B1056 (M)

**Figure 1  Comparison of hominoid pelvis morphology.** Laser scan depictions of left, ventrally oriented ape os coxae for individual specimens of (A) *Hylobates*, (B) *Pongo*, (C) *Gorilla*, and (D) *Pan*. Red arrow indicates the iliopectineal tubercle. Figures not to scale. 

*Owen, 1835*), and others postulating that it may be a highly developed muscle attachment point for several thigh adductors (e.g., *Fick, 1895*). *Owen (1835)* writes that the "spine of the *os pubis* is well marked, but at a greater distance from the symphysis than in the human subject," with the implication that it is indeed the medial attachment point for the inguinal ligament. However, *Sonntag (1924)* describes the orangutan inguinal ligament as being a feeble structure, receiving contributions only from the external oblique aponeurosis but not the other abdominal muscles, and thereby not likely to be associated with such a distinct tubercle. Other researchers have specifically discussed the non-muscular soft tissue structures that attach along the pelvic brim in primates (*Howell & Straus, 1933*; *Miller, 1947*; *Lunn, 1948*; *Trevor-Jones & Van der Spuy, 1970*), though few directly address the tubercle in orangutans. *Miller (1947)* notes a well-developed superior crus of the inguinal ring attaching along the crest of the pubis in the orangutan, although the illustration of this feature indicates that it would attach to the pubis too far medially to come in contact with the more laterally positioned tubercle. *Miller (1947)* additionally makes no mention of a large tubercle on the pubis, suggesting it was not observed in their specimen or not deemed important to mention. These past works cast some uncertainty about whether the large and laterally-positioned tubercle in the iliopectineal region is anatomically homologous to the pubic tubercle in humans.

If this tubercle is not the attachment site of the inguinal ligament and is therefore not anatomically or evolutionarily homologous to the pubic tubercle, there may be other soft tissue structures attached to this bony projection. Trunk muscles (i.e., external oblique, internal oblique, transversus abdominis, rectus abdominis, pyramidalis, psoas minor) and medial thigh muscles (i.e., pectineus, adductor longus, adductor brevis, adductor magnus) have attachments along the pelvic brim and could be associated with this tubercle (see Fig. 3). Aside from the psoas minor and adductor longus (discussed below), most descriptions of primate abdominal and thigh musculature provide little

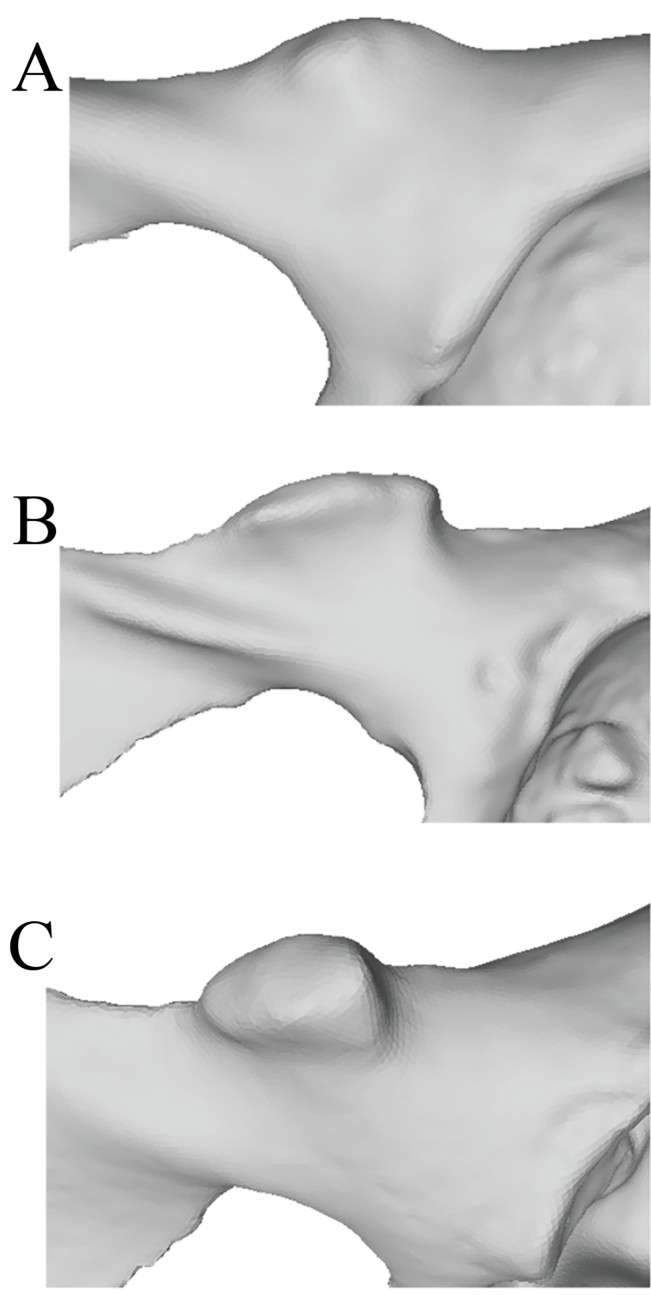

**Figure 2 Laser scan depictions of the iliopectineal tubercle in three *Pongo* specimens.** (A) *P. abelii* (NMNH 143590, adult male), (B) *P. pygmaeus* (NMNH 145305, adult male), and (C) *P. pygmaeus* (NMNH 153823, adult male).               

evidence that any of these muscles would be associated with a large, cranially-projecting tubercle (*Hepburn, 1892*; *Primrose, 1899*; *Sonntag, 1924*; *Boyer, 1935*; *Sigmon, 1974*; *Ferrero et al., 2012*; *Diogo et al., 2013*).

Several muscles are candidates to be associated with the tubercle—herein referred to as the iliopectineal tubercle—based on known function and point of origin/insertion. The primate psoas minor muscle is generally configured similarly to that in humans, coursing distally with the psoas major and inserting along the pelvic brim medial to where the

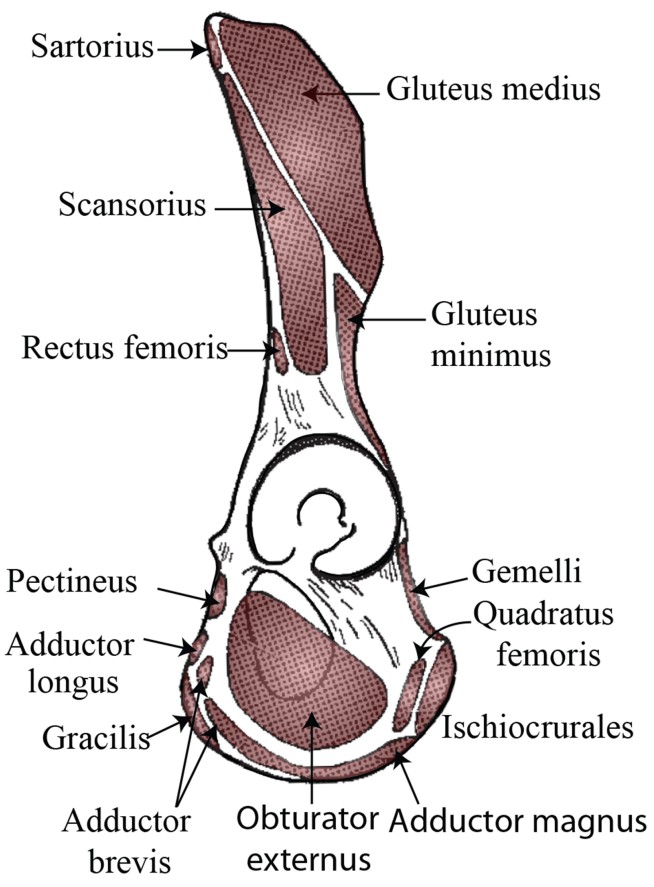

**Figure 3 Muscle map (lateral view) showing historical interpretations of muscle attachment in *Pongo*.** Modified from *Uhlmann (1968)*.

combined iliopsoas tendon runs (*Champneys, 1871*; *Beddard, 1893*; *Bardeen, 1906*; *Lunn, 1948*; *Sigmon, 1974*; *Swindler & Wood, 1973*), though several sources suggest that the orangutan psoas minor inserts on the iliopubic eminence[1], i.e., a slight projection or rugose area of bone where the ilium meets the pubis (*Hepburn, 1892*; *Sigmon, 1974*; *Ferrero et al., 2012*; *Diogo et al., 2013*). The pectineal line is also cited by some researchers as the psoas minor distal attachment (*Hepburn, 1892*; *Fick, 1895*; *Sonntag, 1924*; *Sigmon, 1974*; *Ferrero et al., 2012*; *Diogo et al., 2013*), as is the brim of the pelvis just lateral to the origin of the pectineus (*Primrose, 1899*; *Boyer, 1935*). *Ferrero et al. (2012)* report that the psoas minor inserts on both the femur and the pelvis, though they do not specify beyond that. Although a femoral attachment has never been described by other authors, it is possible that the psoas minor may have inserted into the iliopsoas fascia as has been observed in some human cadavers (*Neumann & Garceau, 2015*), therefore giving a route for insertion onto the lesser trochanter. In his orangutan dissection monograph, *Sonntag (1924)* states that the psoas minor inserts via a powerful tendon onto the posterior part of the iliopectineal line, and that the tendon fans out as it approaches the insertion on the pelvis, communicating some tendinous fibers with the fascia of the psoas major muscle. *Hepburn (1892)* and *Sonntag's (1924)* comparative dissections describe the psoas minor insertion as "further back" (i.e., more laterally/dorsally) on the iliopectineal line in the

[1] It is possible that some of these authors (*Hepburn, 1892*; *Sigmon, 1974*; *Ferrero et al., 2012*; *Diogo et al., 2013*) are using the iliopubic eminence for what we have termed the iliopectineal tubercle, although the iliopubic eminence is not typically considered a bony projection.

orangutan and gorilla than in the chimpanzee. Regrettably, the psoas minor is absent from most anatomical illustrations detailing muscular insertion in the orangutan pelvis (*Thompson, 1901*; *Waterman, 1929*; *Uhlmann, 1968*) despite being illustrated for other hominoids. The only available published work that shows the psoas minor insertion on the orangutan pelvis is *Sigmon (1974)*, which illustrates the psoas minor inserting just posterior to the iliopectineal tubercle, although much of the bony detail in the black and white photographic plate is lacking.

Another potentially viable explanation for the iliopectineal tubercle is as an attachment point for a muscle of the adductor compartment, particularly the adductor longus, adductor brevis, and/or pectineus muscles. In all anthropoid nonhuman primates, the adductor longus originates on the superior pubic ramus and inserts on the femoral shaft, typically on the linea aspera (when present) (*Ferrero et al., 2012*). The proximal origin of the adductor longus in *Pongo* is described as broad, flattened and fleshy compared to the chimpanzee (*Hepburn, 1892*). Concerned primarily with description of *Pongo* musculature, *Fick (1895)* reports that the orangutan adductor longus originates from a bony enthesis he dubs the iliopectineal tubercle, writing that, "The adductor longus muscle arises from a strongly developed iliopectineal tubercle, and is to be described as slender, as in humans. It is supported both medially and laterally by the split pectineus muscle"[2]. The configuration noted by *Fick (1895)* is the first known description of an association of the adductor muscle with the tubercle, but has not been secondarily reported by any others. Some researchers do note that adductor longus muscle originates near the pubic tubercle (*Gibbs, 1999*; *Ferrero et al., 2012*) but do not corroborate the presence of a large tubercle as an attachment point. Additionally, some published figures of the orangutan adductor longus show an origin that would be positionally consistent with the location of the iliopectineal tubercle (*Sigmon, 1974*). The adductor brevis is described as arising from the inferior pubic ramus in all hominids, but not hylobatids (*Ferrero et al., 2012*), though there is no mention of an association with the iliopectineal tubercle aside from *Fick (1895)*. The pectineus is likewise reported as arising from the superior pubic ramus in all apes (*Boyer, 1935*), and also the "pubic tubercle" in *Pongo* (*Ferrero et al., 2012*), though it is again unclear if the authors are referring to the attachment point for the inguinal ligament/external abdominal oblique or the iliopectineal tubercle of *Fick (1895)*.

Although occasionally figured in depictions of orangutan skeletal anatomy (*Thompson, 1901*; *Waterman, 1929*; *Uhlmann, 1968*), the relative prevalence, frequency, and morphological variation of the iliopectineal tubercle has not been systematically characterized in *Pongo* or other primates. Moreover, few comparative soft tissue studies in hominoids have explicitly remarked upon this bony feature when describing soft tissue attachments. As such, it remains unclear what soft tissue element the iliopectineal tubercle is associated with. If the tubercle of the orangutan iliopubic region is truly homologous with the pubic tubercle in humans, it is the attachment point for the inguinal ligament. This may be unlikely given the commonly observed position and robusticity of the iliopectineal tubercle in *Pongo*, and the relatively underdeveloped nature of the inguinal ligament in non-human primates (*Miller, 1947*; *Lunn, 1948*). However, if this feature is

[2] "*Der M adductor longus entspringt von dem starken Tuberculum ileopectineum, ist als schmächtig zu bezeichnen, inserirt wie beim Menschen. Er wird medial und lateral unter lagert von dem gespaltenen M pectineus.*"

associated with another soft tissue, such as a muscle, it may have a significant functional role in orangutan locomotion.

Given the historical confusion over the presence and function of the large tubercle in *Pongo*, we here investigate the morphology and frequency of the iliopectineal tubercle in a comparative context and perform dissections of the soft tissues along the pelvic brim. We assess (1) what soft tissue structure(s) the orangutan iliopectineal tubercle is related to, and (2) whether the iliopectineal tubercle of the orangutan pelvis is a unique character among primates and within Hominoidea. We consider what the functional implications are for the presence of the tubercle in orangutans and discuss its possible evolutionary origin.

## MATERIALS AND METHODS

Cadaveric dissections were performed on a small sample of hominoids to better characterize the soft tissue insertions along the pelvic brim, in the region where the iliopectineal tubercle is found in orangutans (Table 1). These dissections were conducted to specifically test the association of the soft tissues and the iliopectineal tubercle. A standardized anterior route dissection through the abdominal cavity was performed, whereby the abdominal muscles were resected in the midline and any abdominal viscera present were mobilized to expose the iliopsoas complex and pelvic rim. The posterior peritoneum was cleared from the ventral portion of the posterior body wall and the muscles were freed of fascial attachments to better clarify their bony attachment points. The attachment points of the psoas minor were noted, and the attachment point of the distal tendon was cleared of surrounding fascia. To test the potential association of the iliopectineal tubercle with muscles from the adductor compartment of the thigh, the medial portion of the skin and fascia lata were removed and the muscles exposed. The pelvic brim was cleared of fascial attachments and the tubercle (if present) was located via palpation prior to deeper dissection. Neurovascular structures were cleared from the dissection field for visualization purposes. Each of the adductor compartment muscles was isolated and dissected to expose the point of proximal attachment on the pelvis. The cadaveric sample included *Pongo* (n = 4), *Gorilla* (n = 1), *Pan troglodytes* (n = 1), *Pan paniscus* (n = 2), *Symphalangus* (n = 1), and *Hylobates* (n = 3). Additional information about the cadaver sample can be found in Table S1.

We conducted an osteological survey to assess the presence and morphology of the iliopectineal tubercle in a broad sample of primates. Skeletal pelves of n = 294 adult primates from 28 genera were examined for this study (Table 2). Additional data about the osteological sample can be found in Table S2. Genus *Homo* was not included in either our soft tissue or osteological surveys, as human pelvic morphology is well-documented and only presents with a true pubic tubercle as the attachment for the inguinal ligament.

Specimens are from collections at the American Museum of Natural History, the United States National Museum, Cleveland Museum of Natural History, Royal Museum for Central Africa, Bavarian State Collection of Zoology, the Icahn School of Medicine at Mt. Sinai, Zoological *Museum* Amsterdam, Swedish Museum of Natural History, Royal Belgian Institute of Natural Sciences, the University of Chicago, Howard University,

**Table 1 Genera and number of specimens dissected for this study.**

| Genus | Total | Adults | Subadults |
|-------|-------|--------|-----------|
| *Gorilla* | 1 | 1 | 0 |
| *Hylobates* | 3 | 3 | 0 |
| *Pan* | 3 | 3 | 0 |
| *Pongo* | 4 | 4 | 0 |
| *Symphalangus* | 1 | 1 | 0 |

**Note:**
See Table S1 for specimen details.

**Table 2 Genera and number of dry (i.e., osteological) pelvis specimens inspected for this study.**

| Genus | Total | Adults | Subadults |
|-------|-------|--------|-----------|
| *Alouatta* | 9 | 9 | 0 |
| *Arctocebus* | 3 | 3 | 0 |
| *Ateles* | 8 | 8 | 0 |
| *Avahi* | 2 | 2 | 0 |
| *Colobus* | 1 | 1 | 0 |
| *Galago* | 8 | 8 | 0 |
| *Gorilla* | 64 | 62 | 2 |
| *Hoolock* | 23 | 23 | 0 |
| *Hylobates* | 27 | 27 | 0 |
| *Lemur* | 1 | 1 | 0 |
| *Lophocebus* | 2 | 2 | 0 |
| *Loris* | 7 | 7 | 0 |
| *Macaca* | 1 | 1 | 0 |
| *Mandrillus* | 5 | 5 | 0 |
| *Nasalis* | 15 | 15 | 0 |
| *Nomascus* | 3 | 3 | 0 |
| *Nycticebus* | 10 | 10 | 0 |
| *Pan* | 30 | 25 | 5 |
| *Papio* | 9 | 9 | 0 |
| *Perodicticus* | 6 | 6 | 0 |
| *Pithecia* | 2 | 2 | 0 |
| *Pongo* | 51 | 34 | 17 |
| *Propithecus* | 1 | 1 | 0 |
| *Saimiri* | 6 | 6 | 0 |
| *Semnopithecus* | 1 | 1 | 0 |
| *Symphalangus* | 12 | 12 | 0 |
| *Tarsius* | 4 | 4 | 0 |
| *Theropithecus* | 2 | 2 | 0 |

**Note:**
See Table S2 for specimen details.

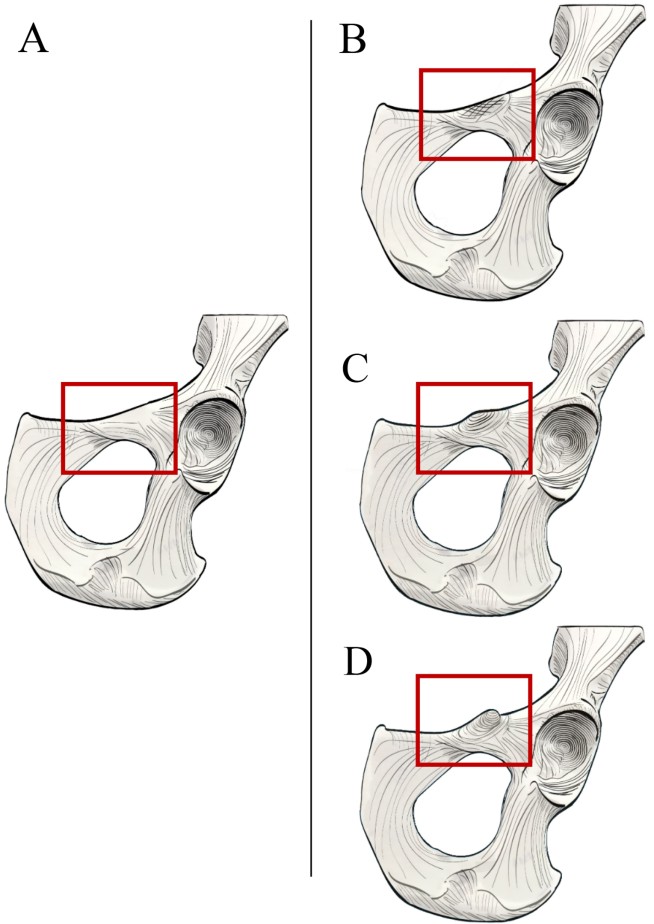

**Figure 4 Illustrated character states for Character 1 (Iliopectineal tubercle presence).** (A) Character state 0 = No tubercle of any size present. Character state 1 = (B) A tubercle of incipient or small, (C) medium, or (D) large size found present.  

Harvard's Museum of Comparative Zoology, the National Museum of Scotland, and the University of Antwerp and Zoo Antwerpen through the Bonobo Morphology Initiative.

## Character definitions

In order to systematically assess the presence and morphology of an iliopectineal tubercle in *Pongo* and other primates, we constructed two qualitative characters to aid in precise identification on osteological specimens. We assessed the iliopectineal tubercle as both a neomorphic (present/absent) and transformational character (multiple character states of varying size) following *Sereno (2007)*, wherein the transformational character was only scored if the neomorphic character was scored as present. As such, all taxa listed above and in Tables 1 and 2 were examined and scored for at least one character, with the second being contingent on the presence of the first. When assessing individual specimens, the neomorphic Character 1 is defined as "Presence of an iliopectineal tubercle (tubercle medial to the anterior inferior iliac spine but lateral to the pectineal line)" and contains the character states: (0) Absent, (1) Present (see Fig. 4). When considering taxon averages, a character state of "(1) Polymorphic" is added for Character 1 as an

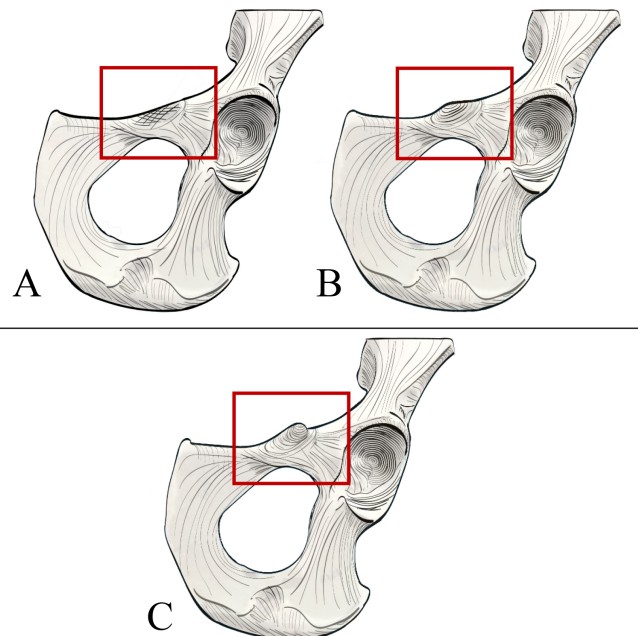

**Figure 5 Illustrated character states for Character 2 (Iliopectineal tubercle size).** Character 2 is only to be scored if Character 1 is marked as present. (A) Character state 0 = Tubercle incipient (increased area of rugosity) or small. (B) Character state 1 = Tubercle is small, with a height less than 1/3$^{rd}$ the size of the superior pubic ramus. (C) Character state 2 = Tubercle is large, with a height greater than 1/3$^{rd}$ the size of the superior pubic ramus.

intermediate state between character states "(0) Absent" and "(2) Present." This character is designed to capture the presence of the tubercle regardless of size or morphology, with the polymorphic state suggesting an intermediate stage where it is not present in all individuals of a taxon. We advocate for a character designation of "(1) Polymorphic" if a tubercle is present in >15% of a population, following *Strait, Grine & Moniz (1997)*. A small, incipient tubercle or area of increased rugosity in the anatomically homologous area should be scored as "(1) Present" when considering an individual specimen, as should a medium or large tubercle.

The transformational Character 2 is defined as "Size of iliopectineal tubercle if scored as present or polymorphic in Character 1" and contains the three character states: (0) An incipient (present as an area of increased rugosity) or small iliopectineal tubercle is present on the iliopectineal eminence medial to the anterior inferior iliac spine and lateral to the pectineal line, (1) Iliopectineal tubercle is present and slightly projecting, with visible rugosity, and a size less than 1/3$^{rd}$ the height of the superior pubic ramus, (2) Iliopectineal tubercle is present and strongly projecting, with total tubercle height at least 1/3$^{rd}$ the height of the superior pubic ramus (see Fig. 5). This character is designed to qualitatively assess the relative size of the tubercle where scored as present by Character 1, and therefore should be based on the relative size within a taxon, not absolute tubercle size. If Character 1 is scored as absent, Character 2 should be scored as "?". If used in a phylogenetic analysis, these characters should be scored as Ordered, as passing from no tubercle to a tubercle, and from a lower character state (i.e., small tubercle) to a higher state

(large tubercle), would logically require passing through intermediate size stages. See Fig. 5 for visual representation. Additionally, where Character 1 is scored as polymorphic or present, we recommend using the modal average character score for Character 2 in a data matrix or a frequency-based coding system for further refinement of results as advocated for morphological systematics by *Wiens (1995)*.

## RESULTS

### Soft tissue survey

Our dissections revealed that the orangutan iliopectineal tubercle is only directly associated with the proximal attachment of the adductor longus muscle, and not with the psoas minor, other adductor muscles, inguinal ligament, or any abdominal muscles that have occasionally been reported as anchoring to the tubercle (see Fig. 6). The proximal tendon of the adductor longus surrounds the iliopectineal tubercle, though its strongest anchoring point is directly on the inferior surface in the midline of the tubercle. The psoas minor distal tendon does not insert directly on the iliopectineal tubercle but is associated with a separate rugose and slightly hypertrophied area of bone along the pelvic brim similar to the condition seen in some quadrupedal non-primate mammals (*Spoor & Badoux, 1988*). The pectineus inserts medial to the iliopectineal tubercle, into a slightly enlarged ridge along the caudoventral aspect of the pubic brim and does not split the adductor longus in our specimens as reported by *Fick (1895)*, who may have observed an anatomical variant in his specimen, though *Hepburn (1892)* also notes a close association between the muscles. The adductor brevis attaches lateral to the iliopectineal tubercle on a slightly raised ridge along the pubic brim that is continuous with the tubercle but does not have a direct attachment point on it. The exclusive attachment of the adductor longus with the iliopectineal tubercle makes it the most likely candidate for causing the entheseal hypertrophy in orangutans, though the pectineus and adductor brevis may contribute by, respectively, building a medial and lateral ridge on the ventral surface of the pubic brim on which the tubercle can form. Our dissections of other hominoids did not reveal an iliopectineal tubercle in any specimen, any association between the iliopectineal tubercle and the adductor longus, or any other muscle of the adductor compartment.

### Osteological survey

Our osteological survey confirmed that the iliopectineal tubercle is a feature consistently present in orangutans, as it was observed in all dry adult (34/34) and most juvenile (14/17) orangutan pelves sampled for this study. The orangutan iliopectineal tubercle is situated along the pectineal line, lateral relative to where the pubic tubercle in humans would be found, confirming the findings of *Fick (1895)*. All adult orangutan osteological specimens surveyed ($n = 34$) had large, projecting iliopectineal tubercles with a modal character score average of 2. Several tubercle morphologies were apparent within the sample of orangutans, though no consistent pattern was noted therein (see Fig. 2). No significant difference was noted in character state distribution by sex, with $n = 15$ males, $n = 15$ females, and $n = 4$ undetermined specimens all exhibiting present and hypertrophied tubercles. The other extant apes exhibited the tubercle in low-to-moderate polymorphic frequencies.

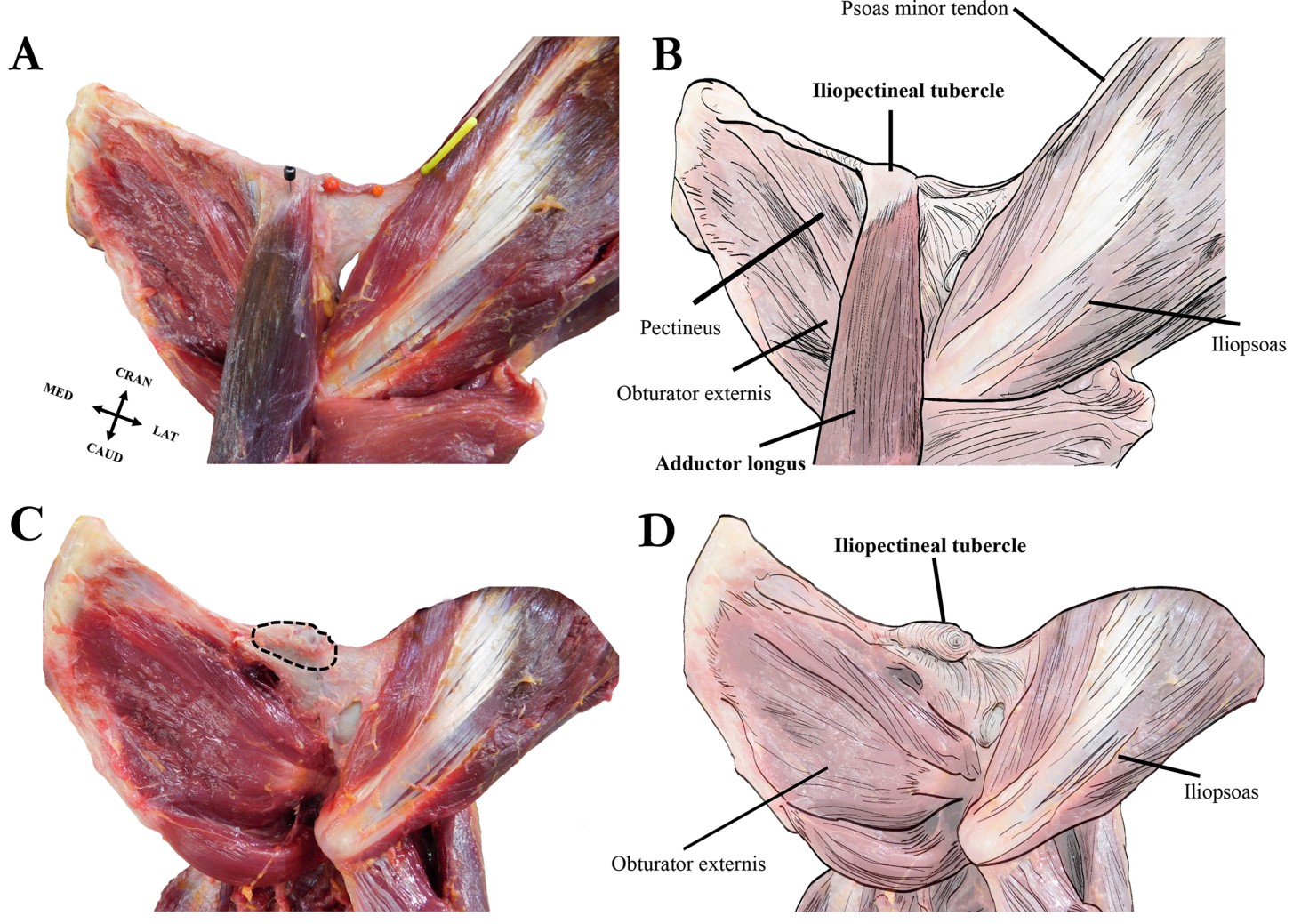

**Figure 6** Left extrinsic pelvic and hindlimb musculature of an adult female Sumatran orangutan (*Pongo abelii*, NMS GH 84.17), anterior view. (A) Photograph of partially dissected musculature, anterior view. Adductor brevis muscle removed. Black pin in proximal attachment of the adductor longus muscle, yellow pin in distal attachment of psoas minor tendon. (B) Illustration of extrinsic pelvic musculature. (C) Photograph of deeper dissection on same specimen. Adductor longus, adductor brevis, and pectineus muscles removed. Iliopectineal tubercle/proximal attachment of adductor longus muscle outlined in dashed black line. (D) Illustration of pelvic musculature with adductor longus, adductor brevis, and pectineus muscles removed.

However, no taxon consistently exhibited a frequency of tubercle presence (Character 1, Character state 1) higher than 52% or a modal character state for Character 2 higher than a score of 1, making the consistency and size of the tubercle in orangutans exceptional within Hominoidea. No osteological specimen observed here exhibited a tubercle either relatively or absolutely as large as that seen commonly in orangutans. No sampled strepsirrhine, platyrrhine, or non-ape catarrhine pelvis exhibited an iliopectineal tubercle that could be considered homologous with that of the orangutan. The majority (14/17) of subadult orangutans observed for this study also exhibit at least an incipient tubercle, while no other juvenile primate exhibited any equivalent tubercle. Frequency data for character presence by genus are tabulated in Table 3 for adults and Table 4 for juveniles.

**Table 3 Character score sums and frequency for all adult specimens by grouped by genus.**

| Genus | n | Character score count | | | | | Character score frequency | | | | |
| | | Character 1 | | Character 2 | | | Character 1 | | Character 2 | | |
| | | C1 = 0 | C1 = 1 | C2 = 0 | C2 = 1 | C2 = 2 | C1 = 0 | C1 = 1 | C2 = 0 | C2 = 1 | C2 = 2 |
|---|---|---|---|---|---|---|---|---|---|---|---|
| *Alouatta* | 9 | 9 | 0 | 0 | 0 | 0 | 1 | 0 | – | – | – |
| *Arctocebus* | 3 | 3 | 0 | 0 | 0 | 0 | 1 | 0 | – | – | – |
| *Ateles* | 8 | 8 | 0 | 0 | 0 | 0 | 1 | 0 | – | – | – |
| *Avahi* | 2 | 2 | 0 | 0 | 0 | 0 | 1 | 0 | – | – | – |
| *Colobus* | 1 | 1 | 0 | 0 | 0 | 0 | 1 | 0 | – | – | – |
| *Galago* | 8 | 8 | 0 | 0 | 0 | 0 | 1 | 0 | – | – | – |
| *Gorilla* | 62 | 30 | 32 | 0 | 32 | 0 | 0.48 | 0.52 | 0 | 1 | 0 |
| *Hoolock* | 23 | 13 | 10 | 6 | 4 | 0 | 0.57 | 0.43 | 0.60 | 0.40 | 0 |
| *Hylobates* | 27 | 24 | 3 | 2 | 0 | 1 | 0.89 | 0.11 | 0.67 | 0 | 0.33 |
| *Lemur* | 1 | 1 | 0 | 0 | 0 | 0 | 1 | 0 | – | – | – |
| *Lophocebus* | 2 | 2 | 0 | 0 | 0 | 0 | 1 | 0 | – | – | – |
| *Loris* | 7 | 7 | 0 | 0 | 0 | 0 | 1 | 0 | – | – | – |
| *Macaca* | 1 | 1 | 0 | 0 | 0 | 0 | 1 | 0 | – | – | – |
| *Mandrillus* | 5 | 5 | 0 | 0 | 0 | 0 | 1 | 0 | – | – | – |
| *Nasalis* | 15 | 15 | 0 | 0 | 0 | 0 | 1 | 0 | – | – | – |
| *Nomascus* | 3 | 1 | 2 | 1 | 1 | 0 | 0.33 | 0.67 | 0.50 | 0.50 | 0 |
| *Nycticebus* | 10 | 10 | 0 | 0 | 0 | 0 | 1 | 0 | – | – | – |
| *Pan* | 25 | 15 | 10 | 2 | 8 | 0 | 0.60 | 0.40 | 0.20 | 0.80 | 0 |
| *Papio* | 8 | 8 | 0 | 0 | 0 | 0 | 1 | 0 | – | – | – |
| *Perodicticus* | 6 | 6 | 0 | 0 | 0 | 0 | 1 | 0 | – | – | – |
| *Pithecia* | 2 | 2 | 0 | 0 | 0 | 0 | 1 | 0 | – | – | – |
| *Pongo* | 34 | 0 | 34 | 2 | 6 | 26 | 0 | 1 | 0 | 0.18 | 0.76 |
| *Propithecus* | 1 | 1 | 0 | 0 | 0 | 0 | 1 | 0 | – | – | – |
| *Saimiri* | 6 | 6 | 0 | 0 | 0 | 0 | 1 | 0 | – | – | – |
| *Semnopithecus* | 1 | 1 | 0 | 0 | 0 | 0 | 1 | 0 | – | – | – |
| *Symphalangus* | 12 | 11 | 1 | 0 | 1 | 0 | 0.92 | 0.08 | 0 | 1 | 0 |
| *Tarsius* | 4 | 4 | 0 | 0 | 0 | 0 | 1 | 0 | – | – | – |
| *Theropithecus* | 2 | 2 | 0 | 0 | 0 | 0 | 1 | 0 | – | – | – |

**Note:**
The – symbol indicates that Character 1 was scored as absent (Character 1, Character state 0) for all specimens in a taxon, and therefore Character 2 could not be scored.

**Table 4 Character score sums and frequency for all subadult specimens grouped by genus.**

| Genus | n | Character score count | | | | | Character score frequency | | | | |
| | | Character 1 | | Character 2 | | | Character 1 | | Character 2 | | |
| | | C1 = 0 | C1 = 1 | C2 = 0 | C2 = 1 | C2 = 2 | C1 = 0 | C1 = 1 | C2 = 0 | C2 = 1 | C2 = 2 |
|---|---|---|---|---|---|---|---|---|---|---|---|
| *Gorilla* | 2 | 2 | 0 | 0 | 0 | 0 | 1 | 0 | – | – | – |
| *Pan* | 5 | 5 | 0 | 0 | 0 | 0 | 1 | 0 | – | – | – |
| *Pongo* | 17 | 3 | 14 | 9 | 4 | 1 | 0 | 0.82 | 0.64 | 0.29 | 0.07 |

**Note:**
The – symbol indicates that Character 1 was scored as absent (Character 1, Character state 0) for all specimens in a taxon, and therefore Character 2 could not be scored.

## DISCUSSION

This study provides two updates to our understanding of *Pongo* anatomy and clarifies contradictions in the historical literature. Our dissections show that the tubercle in orangutans is exclusively associated with the adductor longus, making it clearly not anatomically or developmentally homologous with the pubic tubercle as the medial attachment point for the inguinal ligament. The orangutan pubic tubercle exists as a separate landmark and is associated with a weakly formed inguinal ligament. We also find that a hypertrophied iliopectineal tubercle is a consistent feature observed on the iliopubic eminence of orangutans. This tubercle is additionally present in polymorphic frequencies and smaller sizes in the other hominoids but is not found in any other primate taxon observed here. The presence of a tubercle in all hominoids at a taxon-level frequency above zero to the exclusion of other primate clades suggests that this feature could be an ape synapomorphy, though this idea remains to be formally tested. The soft tissue association remains unclear for the tubercle in other hominoids, as no specimens dissected for this project exhibited a connection between the adductor longus and an iliopectineal tubercle as seen in our orangutans. Further dissections must be conducted to confirm its anatomical homology so that phylogenetic analyses can be performed to determine its evolution within Hominoidea.

Regarding our osteological survey, though variably present in the other hominoids, the iliopectineal tubercle is present at both a higher frequency and a higher average character state in *Pongo* (both *Pongo pygmaeus* and *Pongo abelii*) than in any other primate taxon. Adult orangutans observed here always possess a hypertrophied tubercle, and the majority of juveniles (14/17) observed here possessed at least an incipient tubercle. This suggests that the tubercle's development begins early and continues through adulthood in *Pongo*, where perhaps the tubercle in other apes does not begin development until later in life. Though its growth trajectory is not known, no similarly sized tubercle was observed on juveniles in our sample for any other ape taxon. Taken together with our data on size and character state distribution, our data demonstrate a more consistent and earlier developmental origin for the tubercle in *Pongo* when compared to the other apes. This earlier origin could in part be the result of functional differences in muscle use, earlier differentiation of locomotor specializations, a longer developmental cline related to the slow life-history of orangutans (*Pontzer et al., 2010*), or perhaps some combination therein. Though the developmental pattern of the iliopectineal tubercle has not been systematically documented through all age classes, a relatively earlier ontogenetic origin of the tubercle and/or an earlier emphasis on adductor longus use, paired with the long developmental period of *Pongo* and highly arboreal lifestyle, may create a positive feedback loop encouraging early and consistent iliopectineal tubercle hypertrophy. This may ultimately lead to a more massive adductor longus enthesis throughout all life stages in orangutans. The tubercle's size and consistent presence through all observed growth stages may be a unique feature of *Pongo*, though a larger sample size of juveniles will be needed to robustly assess this hypothesis.

### Character evolution

Given the distribution of the character within our sample, we suggest that there are several potential scenarios for the evolution of the iliopectineal tubercle in extant primates. As the

tubercle is only commonly seen within extant apes we can reduce our search space to Hominoidea to clarify its distribution and polarity on the well-established hominoid phylogenetic tree. In one possible scenario, the presence of an iliopectineal tubercle in at least polymorphic rates is a synapomorphy of Hominidae gained at some point after the split between the apes and their most recent common ancestor with the cercopithecoid monkeys ~30 Mya (*Arnold, Matthews & Nunn, 2010*). In this scenario, the character is maintained in at least polymorphic frequencies in *Pongo*, *Gorilla*, *Pan*, and the hylobatids, but is lost in *Homo* at some point after the split from the most recent common ancestor with *Pan*. A scenario invoking homoplasy must also be considered, with the iliopectineal tubercle potentially evolving multiple times within the apes. This would necessitate that the character evolved to be present in polymorphic frequencies at least five times within crown Hominoidea, though this scenario cannot be currently excluded from consideration as the mechanisms for growth, development, and maintenance are currently unknown.

Based on our osteological assessment, we suggest that the iliopectineal tubercle presence (Character 1) has moved towards fixation in *Pongo*, while the other apes exhibit it at only moderate polymorphic frequencies. Regarding the evolution of Character 2 (size), *Pongo* has moved toward fixation of Character state 2 in adults, where the iliopectineal tubercle is greatly hypertrophied. As the large tubercle is seen in equal distribution in both *Pongo pygmaeus* and *Pongo abelii*, the most parsimonious inference is that its current morphology was present in the pongid ancestor prior to the species divergence of Sumatran and Bornean orangutan lineages some ~4 Mya (*Raaum et al., 2005*; *Chatterjee et al., 2009*; *Arnold, Matthews & Nunn, 2010*; *Pozzi et al., 2014*). The selection for an enlarged tubercle likely occurred at some point along the lineage leading to extant orangutans but prior to the divergence of the extant taxa, barring less parsimonious explanations involving independent hypertrophy. Discovery of fossil taxa with this trait preserved may further elucidate its origin point and evolutionary history within the apes.

The functional implications of a large tubercle for the adductor longus muscle are not explicitly clear. As with other muscle attachment points, a large, projecting enthesis could provide a greater muscle moment arm and larger mechanical advantage for the adductor longus (*Karakostis & Lorenzo, 2016*). One possible functional explanation for the presence of an iliopectineal tubercle is that it is an adaptation for increased arboreality at a large body size, given that it is prevalent in orangutans and more frequently observed in gorillas than in chimpanzees or the hylobatids (Table 3). However, if arboreality were the prime driver of tubercle development regardless of taxon, we would expect to see the tubercle at a lower frequency in the eastern gorilla (*Gorilla beringei*) when compared to the significantly more arboreal Western gorilla (*G. gorilla*). This hypothesis is not borne out by our study sample, in which all gorilla species and subspecies had identical modal character scores. If the tubercle size and frequency were purely a function of body mass, we would expect it to be largest and most common in the largest animals sampled here (i.e., gorillas). This prediction is also not borne out by our data, which demonstrate that *Pongo* consistently exhibits a larger iliopectineal tubercle than the absolutely more massive *Gorilla*. From this we infer that tubercle size is not strictly related to either arboreality or body mass but likely some combination of the two, and perhaps

to unique modes of acrobatic locomotion in orangutans. It is worth noting here, though, that additional ape dissections are necessary to establish whether the tubercle is truly homologous in the hominoids, as none of our ape soft tissue specimens of any taxon other than *Pongo* exhibited a tubercle in association with the adductor longus muscle.

Direct examination of the adductor longus muscle may be insightful for understanding its effect on the iliopectineal tubercle, but published electromyographic (EMG) data on the adductor longus in primates is limited, further restricting our ability to draw functional inferences. *Stern & Larson (1993)* report that the adductor longus is most active in *Pongo* just prior to the beginning of liftoff for swing-phase when walking bipedally, firing in sequence with the obturator internus and obturator externus to stabilize the thigh. Its function during a more typical mode of orangutan locomotion involving quadrumanous arboreality is not reported but can be surmised based on attachment to perform adduction, medial rotation, and some flexion of the thigh in *Pongo* as in other primates. As such, the hypertrophied tubercle may represent an adaptation for the highly arboreal, quadrumanous locomotion that typifies orangutan movement, allowing for greater adduction forces with less muscular effort. It may also simply be the effect of more frequent use and hypertrophy of the adductor longus muscle throughout life, though studies of hominoid limb muscles demonstrate that the adductor compartment of *Gorilla* is in fact relatively larger than in *Pongo* when compared to body mass, suggesting relative mass may not be an effective proxy for use (*Zihlman, Mcfarland & Underwood, 2011*). Data on the specific weights of adductor compartment muscles are otherwise limited for *Pongo*, as are the specific action of the adductor longus during various phases of typical orangutan locomotion. Further investigation of in vivo locomotion including spanning behaviors, hanging, and quadrumanous acrobatics may increase our understanding of adductor longus muscle use, as no published EMG studies directly assess its function while climbing in complex arboreal habitats.

We see several possible avenues for future research. As already mentioned, additional EMG data would be instrumental in clarifying when and how ape hindlimbs are used during locomotion reflective of normal use (i.e., not only bipedal walking). Data already show that orangutans use highly abducted hip postures and have a large range of hip abduction–adduction (*Hammond, 2014*; *Hammond, Plavcan & Ward, 2016*), and likely have other soft tissue adaptations to facilitate high levels of hip joint mobility (*Hammond, 2016*; *Muchlinski et al., 2018*). It is uncertain when the adductor longus is typically recruited during adduction or in stabilization of the hip in various postures. Further study of the pelvic brim and adductor longus configuration in other primate or mammal groups could also be relevant. It has been reported that slow-moving sloths and slow lorises have long pubic rami, possibly as an adaptation to increase the moment arm for abdominal musculature (*Lewton & Dingwall, 2016*). There may be some increased leverage for the orangutan abdominal wall as well, related to ancillary shape changes of the pelvic brim related to the adductor longus tubercle. Additionally, larger atelines that frequently engage in suspensory behaviors (i.e., *Ateles*, *Lagothrix*, *Brachyteles*) have been shown to converge on ape morphologies (*Larson, 1998*), though our osteological survey found no evidence of an iliopectineal tubercle in a relatively small sample of *Ateles* ($n = 8$).

Future studies of the iliopectineal tubercle and associated musculature would benefit from incorporating these taxa more extensively. Furthermore, though a myriad of research has established how differences in muscle fiber type composition is related to function in primate locomotion (*Hall-Craggs, 1974*; *Sickles & Pinkstaff, 1981*; *Acosta & Roy, 1987*; *Schmidt & Schilling, 2007*; *Huq et al., 2018*), comparative immunohistochemical analyses of the primate adductor muscles are lacking. A significant difference in adductor longus fiber type and muscle architecture among the apes could prove critical for determining if the muscle is unique in *Pongo* in its form or function and may provide data on its influence in forming the iliopectineal tubercle.

Finally, while fossil hominoid pelves are rare, the reinterpretation of this tubercle in the fossil record may have implications for understanding both the systematic relationships and the locomotor capabilities of extinct primates. If, as we discuss as a possibility above, the presence of an iliopectineal tubercle is a hominoid synapomorphy, we should expect to see it in stem hominoid taxa that are possibly sister clades to extant groups. Unfortunately, the region on the superior pubic ramus where a tubercle might present is rarely preserved in published stem hominoid specimens. Regardless, understanding that the iliopectineal tubercle is not an attachment for the inguinal ligament, but rather for the adductor longus muscle, should change functional interpretations if its presence is observed in fossil taxa.

## CONCLUSIONS

This study confirms that a large and projecting tubercle on the pelvic brim is a feature consistently found in orangutans. Cadaveric dissections revealed that this tubercle is specifically associated with the proximal origin of the adductor longus in orangutans and is therefore not likely homologous to the pubic tubercle in form or function, supporting the observations of *Fick (1895)* and refuting other conflicting historical reports. We demonstrate that a similar, though less robust, tubercle is polymorphically observed in the other apes, but is not found in other primate clades observed here. The restricted presence within the extant primate clades observed here suggests that the iliopectineal tubercle is possibly an ape synapomorphy that has been fixed in *Pongo* at a high rate and a consistently hypertrophied state. While the functional role of the large adductor longus tubercle in orangutans is not yet fully understood, we hypothesize that it relates to highly acrobatic locomotion necessitating powerful adduction movements and/or hip stabilization. Finally, though no formal systematic tests were conducted for this study, we encourage researchers to utilize the characters and character states we have described here in future work to further our understanding of orangutan evolution, including the limited fossil record of hominoid pelves.

## ACKNOWLEDGEMENTS

We thank the Department of Anatomical Sciences at Stony Brook University, the Center for Anatomy and Functional Morphology at the Icahn School of Medicine at Mount Sinai, the Department of Anatomy at Howard University, the Center for Research

and Conservation of the Royal Zoological Society of Antwerp and the Bonobo Morphology Initiative, and the National Museum of Scotland for providing the cadaveric materials and/ or providing a comparative context for the authors. We would like to extend personal thanks to Danny Soto, Bonnie Sumner, Susan Larson, Jeffrey Laitman, Joy Reidenberg, Richard Madden, Myra Laird, Rui Diogo, Sandra Nauwelaerts, Jeroen Stevens, Stephen Rodgers, Georg Hantke, Alan Lothain, and Andrew Kitchener for facilitating dissection-based research. Curators and staff at the American Museum of Natural History, Cleveland Museum of Natural History, National Museum of Natural History, and Harvard's Museum of Comparative Zoology are thanked for access to osteological specimens. Thanks to Brigitte Demes and Chris Cantrell for assistance with translation, and to Kelsey Pugh for discussions of ape systematics. We gratefully acknowledge the editor and the three anonymous reviewers for their comments.

### Funding

This work has been supported by the National Science Foundation (No. BCS 1440624) (to Magdalena Muchlinski), the Wenner-Gren Foundation (to Ashley S. Hammond), the L. S. B. Leakey Foundation (to Ashley S. Hammond), and the National Science Foundation Graduate Research Fellowship Program grant (to Brian M. Shearer). The funders had no role in study design, data collection and analysis, decision to publish, or preparation of the manuscript.

### Grant Disclosures

The following grant information was disclosed by the authors:
National Science Foundation: BCS 1440624.
Wenner-Gren Foundation.
L. S. B. Leakey Foundation.
National Science Foundation Graduate Research Fellowship Program.

### Competing Interests

The authors declare that they have no competing interests.

### Author Contributions

- Brian M. Shearer conceived and designed the experiments, performed the experiments, analyzed the data, prepared figures and/or tables, authored or reviewed drafts of the paper, approved the final draft, illustrations.
- Magdalena Muchlinski conceived and designed the experiments, performed the experiments, analyzed the data, prepared figures and/or tables, authored or reviewed drafts of the paper, approved the final draft.
- Ashley S. Hammond conceived and designed the experiments, performed the experiments, analyzed the data, prepared figures and/or tables, authored or reviewed drafts of the paper, approved the final draft.

## Animal Ethics

The following information was supplied relating to ethical approvals (i.e., approving body and any reference numbers):

This project consisted of observation of museum curated skeletal collections and soft tissue specimens. No animals were sacrificed and no animals were kept in captivity for this study. No ethics committees were needed to complete this work.

## Data Availability

Data and character scores for all soft tissue specimens are available in Table S1. Data for all osteological specimens are available in Table S2. Specimens are from collections at the American Museum of Natural History (AMNH), the United States National Museum (USNM), Cleveland Museum of Natural History (CMNH), Royal Museum for Central Africa (RMCA), Bavarian State Collection of Zoology (ZSM), the Icahn School of Medicine at Mt. Sinai (MS), Zoological Museum Amsterdam (ZMA), Swedish Museum of Natural History (NRM), Royal Belgian Institute of Natural Sciences (RBINS), the University of Chicago (UC), Howard University (HU), Harvard's Museum of Comparative Zoology (MCZ), the National Museum of Scotland (NMS), and the University of Antwerp and Zoo Antwerpen through the Bonobo Morphology Initiative (AU). Specimen numbers are available in Tables S1 and S2.

## Supplemental Information

Supplemental information for this article can be found online at http://dx.doi.org/10.7717/peerj.7273#supplemental-information.

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
