# Peer review of "Large pelvic tubercle in orangutans relates to the adductor longus muscle"

_PeerJ, doi:10.7717/peerj.7273_

## Round 0.1 · original submission · Minor Revisions

Your manuscript has been seen by three referees who provided information how to improve it. Please take their comments in full consideration.

Reviewer 1 ·

Basic reporting

No comment

Experimental design

No comment

Validity of the findings

No comment

Additional comments

The presented manuscript provides valuable information in the field of comparative and evolutionary anatomy that is related to the presence and the great development of the iliopectineal tubercle in orangutans, as well as its relationship with the tendon of insertion of the adductor longus muscle. The manuscript is well written and its methodology conforms to the professional standards required in this type of research work. In my opinion, I believe that the publication of the manuscript after a minor revision can be considered.
- In the Materials & Methods section, as well as in Tables 1 and 2, it should be specified if the osteological sample used belongs to individuals raised in the wild or in captivity. In the case of having individuals from both sources it would be interesting to analyze, especially in Pongo, if there are differences between them in the presence and/or in the development of the iliopectineal tubercle. Do the authors think that the development of different locomotion patterns in primates raised in the wild and in captivity can affect the presence and/or development of the iliopectineal tubercle? Also, no information is presented on the number of males and females in the osteological sample analyzed. I think it would be interesting to add this information and analyze the existence of possible differences between males and females to see how the iliopectineal tubercle behaves in highly dimorphic genera such as Pongo and Gorilla.
- I also think it would be interesting to add quantitative information regarding the adductor longus muscle. During the dissections carried out, have the authors collected quantitative muscle information, such as the absolute weight of the muscles, the length of the muscle fascicles and the angle of pennation? With this information you can obtain interesting quantitative parameters such as the absolute and relative weight of the adductor longus with respect to the other muscles of the thigh or with respect to body weight, or calculate the physiological cross-sectional area. With this information, it can be analyzed if there is a relationship between the values of these parameters and the presence and/or development of the iliopectineal tubercle comparing the different dissected species among them. Although the number of dissected individuals is low for some genera (Gorilla), the fact of having dissected 4 Pongo, 3 Pan and 3 Hylobates can provide information on the relationship between the iliopectineal tubercle and the anatomical and functional characteristics of the adductor longus.
- In Figure 4 the muscle designated as psoas major in B and D corresponds at this level to the iliopsoas muscle. In the legend of the image 4c it should be indicated that the pectineus muscle has also been eliminated.
- In Table 3, the value of C2 = 1 in Gorilla should be 32?
- In Table 4, the value of C2 = 0 in Pan should be 0?

Reviewer 2 ·

Basic reporting

The manuscript is very good and meets all the relevant criteria for this section. It is clear and well-written. The introduction and background provide plenty of historical context regarding the interpretation of the iliopectineal tubercle, and clearly communicate why this study is of interest. The figures, especially 2, 3, and 4, are very good. I suppose that space isn’t an issue so it is by no means necessary, but it seems to me that Figures 2 and 3 could be combined into a single figure with no loss of clarity. However, I understand if the authors wish to keep them separate.

I did notice that Fick 1895a is referenced once in the text but is missing from the list of Works Cited.

Experimental design

This research fills a small but important gap in our knowledge about the prevalence and expression of what the authors have termed the iliopectineal tubercle. The research questions are pointed and well-defined. My only comment about the design as it currently stands is that Homo is not included in the osteological sample. If one is going to assess whether the iliopectineal tubercle is unique among primates and within Hominoidea, it seems that Homo should be included, even if we are very confident that no iliopectineal tubercle will be found. If I had to guess why Homo was not included, it would be because one of the stated aims of the study is to determine if the iliopectineal tubercule is homologous to the pubic tubercle in humans and thus there would be an element of circularity in determining the character state in Homo. One solution to this is to “reverse” the research questions as stated in lines 135-137 (and as reported in the results and discussion). First, it is assessed which soft tissue structure the iliopectineal tubercle is related to, then assess whether the tubercle is unique among primates/hominoids. Since it had been determined by the soft tissues that the iliopectineal tubercle is NOT homologous to the pubic tubercle in humans (as was suspected), the authors are then free to score the character in humans (presumably as zero) without interference. Now, I could be completely missing the mark here, and this is by no means an insistence on including modern humans in the osteological sample. But I did wonder why they were not included and it might be worth addressing why you chose not to.

Validity of the findings

The authors find that the iliopectineal tubercule is expressed in all adult and some subadult osteological specimens of Pongo. The authors find the iliopectineal tubercle variably expressed in other adult hominoids (with the exception of Homo), and not at all in other subadults. Dissection shows that, in Pongo, the iliopectineal tubercule is associated with the proximal attachment of the adductor longus and is not homologous with the pubic tubercle of Homo. These findings seem valid and are well supported by the results of their study.

Additional comments

I enjoyed reading the paper, nice work!

Reviewer 3 ·

Basic reporting

The article is well-written and clear, and provides sufficient introduction and background to demonstrate how the work fits into the broader literature. The figures are relevant and clearly illustrated.

Experimental design

The submission clearly defines the research questions in lines 134-137 and places them within a functional and evolutionary context. The methods are clearly described, and the sample is of an appropriate size and composition to address the research questions. The authors note that a larger cadaver sample and one encompassing a broader age range would be useful in future investigations.

Validity of the findings

The data are robust (n = 294, 28 primate genera) and included in the tables and supplemental information. The conclusions are appropriately stated and connected to the original research questions, and supported by the results.

Additional comments

My only general comment has to do with lines 190-192, in which the authors describe the location of the tubercle as medial to the anterior superior iliac spine (“An incipient...or small iliopectineal tubercle is present on the iliopectineal eminence medial to the anterior superior iliac spine and lateral to the pectineal line...”). Given the shape of hominoid ilia, most features on the iliopubic ramus are medial to the ASIS. Can the tubercle's description be clarified with reference to another feature?

---

## Round 0.2 · Minor Revisions

Thank you very much for your consideration of all reviewers'suggestions. I have one more point that I would like you to address.In the Abstract you stressed that this work will clarify "whether this tubercle is a unique anatomical feature of Pongo, or if it is anatomically, functionally, or phylogenetically homologous with the human pubic tubercle when considered as a soft tissue attachment point " However,as you have no performed functional or phylogentic analyses the goals posited do not seem achievable.Please reprhase with a less strong sentence.For example, this work will contribute with anatomical knowledgement to... or something like this.

---

## Round 0.3 · accepted · Accept

Thank you very much, I think that we are ready to move on to production.